# Observational-Interventional Priors for Dose-Response Learning

**Ricardo Silva**
Department of Statistical Science and Centre for Computational Statistics and Machine Learning
University College London
ricardo@stats.ucl.ac.uk

## Abstract

Controlled interventions provide the most direct source of information for learning causal effects. In particular, a dose-response curve can be learned by varying the treatment level and observing the corresponding outcomes. However, interventions can be expensive and time-consuming. Observational data, where the treatment is not controlled by a known mechanism, is sometimes available. Under some strong assumptions, observational data allows for the estimation of dose-response curves. Estimating such curves nonparametrically is hard: sample sizes for controlled interventions may be small, while in the observational case a large number of measured confounders may need to be marginalized. In this paper, we introduce a hierarchical Gaussian process prior that constructs a distribution over the dose-response curve by learning from observational data, and reshapes the distribution with a nonparametric affine transform learned from controlled interventions. This function composition from different sources is shown to speed-up learning, which we demonstrate with a thorough sensitivity analysis and an application to modeling the effect of therapy on cognitive skills of premature infants.

## 1 Contribution

We introduce a new solution to the problem of learning how an outcome variable $Y$ varies under different levels of a control variable $X$ that is manipulated. This is done by coupling different Gaussian process priors that combine observational and interventional data. The method outperforms estimates given by using only observational or only interventional data in a variety of scenarios and provides an alternative way of interpreting related methods in the design of computer experiments.

Many problems in causal inference [14] consist of having a treatment variable $X$ and and outcome $Y$, and estimating how $Y$ varies as we *control* $X$ at different levels. If we have data from a randomized controlled trial, where $X$ and $Y$ are not *confounded*, many standard modeling approaches can be used to learn the relationship between $X$ and $Y$. If $X$ and $Y$ are measured in an *observational study*, the corresponding data can be used to estimate the association between $X$ and $Y$, but this may not be the same as the causal relationship of these two variables because of possible confounders.

To distinguish between the observational regime (where $X$ is not controlled) and the interventional regime (where $X$ is controlled), we adopt the causal graphical framework of [16] and [19]. In Figure 1 we illustrate the different regimes using causal graphical models. We will use $p(\cdot \mid \cdot)$ to denote (conditional) density or probability mass functions. In Figure 1(a) we have the observational, or "natural," regime where common causes $\mathbf{Z}$ generate both treatment variable $X$ and outcome variable $Y$. While the conditional distribution $p(Y = x \mid X = x)$ can be learned from this data, this quantity is not the same as $p(Y = y \mid do(X = x))$: the latter notation, due to Pearl [16], denotes a regime where $X$ is not random, but a quantity set by an intervention performed by an external agent. The relation between these regimes comes from fundamental invariance assumptions: when $X$ is intervened upon,

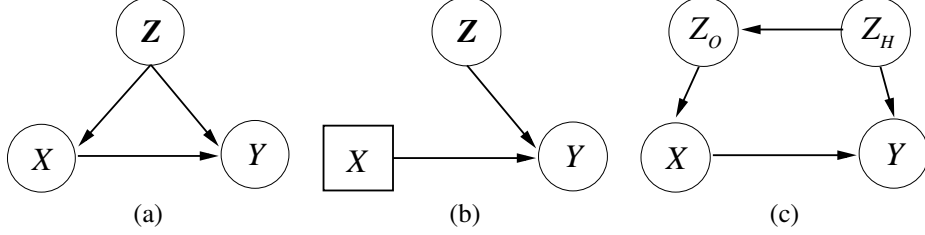

Figure 1: Graphs representing causal graphical models. Circles represent random variables, squares represent fixed constants. (a) A system where $\mathbf{Z}$ is a set of common causes (confounders), common parents of $X$ and $Y$ here represented as a single vertex. (b) An intervention overrides the value of $X$ setting it to some constant. The rest of the system remains invariant. (c) $Z_O$ is not a common cause of $X$ and $Y$, but blocks the influence of confounder $Z_H$.

"all other things are equal," and this invariance is reflected by the fact that the model in Figure 1(a) and Figure 1(b) share the same conditional distribution $p(Y = x|X = x, \mathbf{Z} = \mathbf{z})$ and marginal distribution $p(\mathbf{Z} = \mathbf{z})$. If we observe $\mathbf{Z}$, $p(Y = y \mid do(X = x))$ can be learned from observational data, as we explain in the next section.

Our goal is to learn the relationship

$$f(x) \equiv \mathbb{E}[Y \mid do(X = x)], x \in \mathcal{X}, \qquad (1)$$

where $\mathcal{X} \equiv \{x_1, x_2, \ldots, x_T\}$ is a pre-defined set of treatment levels. We call the vector $f(\mathcal{X}) \equiv [f(x_1); \ldots; f(x_T)]^\top$ the response curve for the "doses" $\mathcal{X}$. Although the term "dose" is typically associated with the medical domain, we adopt here the term *dose-response learning* in its more general setup: estimating the causal effect of a treatment on an outcome across different (quantitative) levels of treatment. We assume that the causal structure information is known, complementing approaches for structure learning [19, 9] by tackling the quantitative side of causal prediction.

In Section 2, we provide the basic notation of our setup. Section 3 describes our model family. Section 4 provides a thorough set of experiments assessing our approach, including sensitivity to model misspecification. We provide final conclusions in Section 5.

## 2 Background

The target estimand $p(Y = y \mid do(X = x))$ can be derived from the structural assumptions of Figure 1(b) by standard conditioning and marginalization operations:

$$p(Y = y \mid do(X = x)) = \int p(Y = y \mid X = x, \mathbf{Z} = \mathbf{z}) p(\mathbf{Z} = \mathbf{z}) \, d\mathbf{z}. \qquad (2)$$

Notice the important difference between the above and $p(Y = y \mid X = x)$, which can be derived from the assumptions in Figure 1(a) by marginalizing over $p(\mathbf{Z} = \mathbf{z} \mid X = x)$ instead. The observational and interventional distributions can be very different. The above formula is sometimes known as the *back-door adjustment* [16] and it does not require measuring all common causes of treatment and outcome. It suffices that we measure variables $\mathbf{Z}$ that block all "back-door paths" between $X$ and $Y$, a role played by $Z_O$ in Figure 1(c). A formal description of which variables $\mathbf{Z}$ will validate (2) is given by [20, 16, 19]. We will assume that the selection of which variables $\mathbf{Z}$ to adjust for has been decided prior to our analysis, although in our experiments in Section 4 we will assess the behavior of our method under model misspecification. Our task is to estimate (1) nonparametrically given observational and experimental data, assuming that $\mathbf{Z}$ satisfies the back-door criteria.

One possibility for estimating (1) from observational data $\mathcal{D}_{obs} \equiv \{(Y^{(i)}, X^{(i)}, \mathbf{Z}^{(i)})\}, 1 \leq i \leq N$, is by first estimating $g(x, \mathbf{z}) \equiv \mathbb{E}[Y \mid X = x, \mathbf{Z} = \mathbf{z}]$. The resulting estimator,

$$\hat{f}(x) \equiv \frac{1}{N} \sum_{i=1}^{N} \hat{g}(x, \mathbf{z}^{(i)}), \qquad (3)$$

is consistent under some general assumptions on $f(\cdot)$ and $g(\cdot, \cdot)$. Estimating $g(\cdot, \cdot)$ nonparametrically seems daunting, since $\mathbf{Z}$ can in principle be high-dimensional. However, as shown by [5], under

some conditions the problem of estimating $\hat{f}(\cdot)$ nonparametrically via (3) is no harder than a one-dimensional nonparametric regression problem. There is however one main catch: while observational data can be used to choose the level of regularization for $\hat{g}(\cdot)$, this is not likely to be an optimal choice for $\hat{f}(\cdot)$ itself. Nevertheless, even if suboptimal smoothing is done, the use of nonparametric methods for estimating causal effects by back-door adjustment has been successful. For instance, [7] uses Bayesian classification and regression trees for this task.

Although of practical use, there are shortcomings in this idea even under the assumption that $\mathbf{Z}$ provides a correct back-door adjustment. In particular, Bayesian measures of uncertainty should be interpreted with care: a fully Bayesian equivalent to (3) would require integrating over a model for $p(\mathbf{Z})$ instead of the empirical distribution for $\mathbf{Z}$ in $\mathcal{D}_{obs}$; evaluating a dose $x$ might require combining many $g(x, \mathbf{z}^{(i)})$ where the corresponding training measurements $x^{(i)}$ are far from $x$, resulting on possibly unreliable extrapolations with poorly calibrated credible intervals. While there are well established approaches to deal with this "lack of overlap" problem in binary treatments or linear responses [18, 8], it is less clear what to do in the continuous case with nonlinear responses.

In this paper, we focus on a setup where it is possible to collect interventional data such that treatments are controlled, but where sample sizes might be limited due to financial and time costs. This is related to design of computer experiments, where (cheap, but biased) computer simulations are combined with field experiments [2, 6]. The key idea of combining two sources of data is very generic, the value of new methods being on the design of adequate prior families. For instance, if computer simulations are noisy, it is may not be clear how uncertainty at that level should be modeled. We leverage knowledge of adjustment techniques for causal inference, so that it provides a partially automated recipe to transform observational data into informed priors. We leverage knowledge of the practical shortcomings of nonparametric adjustment (3) so that, unlike the biased but low variance setup of computer experiments, we try to improve the (theoretically) unbiased but possibly oversmooth structure of such estimators by introducing a layer of pointwise affine transformations.

**Heterogeneous effects and stratification.** One might ask why marginalize $\mathbf{Z}$ in (2), as it might be of greater interest to understand effects at the finer subpopulation levels conditioned on $\mathbf{Z}$. In fact, (2) should be seen as the most general case, where conditioning on a subset of covariates (for instance, gender) will provide the possibly different average causal effect for each given strata (different levels of gender) marginalized over the remaining covariates. Randomized fine-grained effects might be hard to estimate and require stronger smoothing and extrapolation assumptions, but in principle they could be integrated with the approaches discussed here. In practice, in causal inference we are generally interested in marginal effects for some subpopulations where many covariates might not be practically measurable at decision time, and for the scientific purposes of understanding *total effects* [5] at different levels of granularity with weaker assumptions.

## 3 Hierarchical Priors via Inherited Smoothing and Local Affine Changes

The main idea is to first learn from observational data a Gaussian process over dose-response curves, then compose it with a nonlinear transformation biased toward the identity function. The fundamental innovation is the construction a nonstationary covariance function from observational data.

### 3.1 Two-layered Priors for Dose-responses

Given an observational dataset $\mathcal{D}_{obs}$ of size $N$, we fit a Gaussian process to learn a regression model of outcome $Y$ on (uncontrolled) treatment $X$ and covariates $\mathbf{Z}$. A Gaussian likelihood for $Y$ given $X$ and $\mathbf{Z}$ is adopted, with conditional mean $g(x, \mathbf{z})$ and variance $\sigma_g^2$. A Matérn 3/2 covariance function with automatic relevance determination priors is given to $g(\cdot, \cdot)$, followed by marginal maximum likelihood to estimate $\sigma_g^2$ and the covariance hyperparameters [12, 17]. This provides a posterior distribution over functions $g(\cdot, \cdot)$ in the input space of $X$ and $\mathbf{Z}$. We then define $f_{obs}(\mathcal{X})$, $x \in \mathcal{X}$, as

$$f_{obs}(x) \equiv \frac{1}{N} \sum_{i=1}^{N} g(x, \mathbf{z}^{(i)}), \tag{4}$$

where set $\{g(x, \mathbf{z}^{(i)})\}$ is unknown. Uncertainty about $f_{obs}(\cdot)$ comes from the joint predictive distribution of $\{g(x, \mathbf{z}^{(i)})\}$ learned from $\mathcal{D}_{obs}$, itself a Gaussian distribution with a $TN \times 1$ mean vector

$\mu_g^\star$ and a $TN \times TN$ covariance matrix, $T \equiv |\mathcal{X}|$. Since (4) is a linear function of $\{g(x, \mathbf{z}^{(i)})\}$, this implies $f_{obs}(\mathcal{X})$ is also a (nonstationary) Gaussian process with mean $\mu_{obs}(x) = \frac{1}{N} \sum_{i=1}^{N} \mu_g^\star(x, z^{(i)})$ for each $x \in \mathcal{X}$. The motivation for (4) is that $\mu_{obs}$ is an estimator of the type (3), inheriting its desirable properties and caveats.

The cost of computing the covariance matrix $K_{obs}$ of $f_{obs}(\mathcal{X})$ is $\mathcal{O}(T^2 N^2)$, potentially expensive. In many practical applications, however, the size of $\mathcal{X}$ is not particularly large as it is a set of intervention points to be decided according to practical real-world constraints. In our simulations in Section 4, we chose $T = |\mathcal{X}| = 20$. Approximating such covariance matrix, if necessary, is a future research topic.

Assume interventional data $\mathcal{D}_{int} \equiv \{(Y_{int}^{(i)}, x_{int}^{(i)})\}, 1 \le i \le M$, is provided (with assignments $x_{int}^{(i)}$ chosen by some pre-defined design in $\mathcal{X}$). We assign a prior to $f(\cdot)$ according to the model

$$
\begin{aligned}
f_{obs}(\mathcal{X}) &\sim \mathcal{N}(\mu_{obs}, K_{obs}) \\
a(\mathcal{X}) &\sim \mathcal{N}(\mathbf{1}, K_a) \\
b(\mathcal{X}) &\sim \mathcal{N}(\mathbf{0}, K_b) \\
f(\mathcal{X}) &= a(\mathcal{X}) \odot f_{obs}(\mathcal{X}) + b(\mathcal{X}) \\
Y_{int}^{(i)} &\sim \mathcal{N}(f(x_{int}^{(i)}), \sigma_{int}^2), 1 \le i \le M,
\end{aligned}
\tag{5}
$$

where $\mathcal{N}(\mathbf{m}, \mathbf{V})$ is the multivariate normal distribution with mean $\mathbf{m}$ and covariance matrix $\mathbf{V}$, $\odot$ is the elementwise product, $a(\cdot)$ is a vector which we call the *distortion function*, and $b(\cdot)$ the *translation function*. The role of the "elementwise affine" transform $a \odot f_{obs} + b$ is to bias $f$ toward $f_{obs}$ with uncertainty that varies depending on our uncertainty about $f_{obs}$. The multiplicative component $a \odot f_{obs}$ also induces a heavy-tail prior on $f$. In the Supplementary Material, we discuss briefly the alternative of using the deep Gaussian process of [4] in our observational-interventional setup.

## 3.2 Hyperpriors

We parameterize $K_a$ as follows. Every entry $k_a(x, x')$ of $K_a$, $(x, x') \in \mathcal{X} \times \mathcal{X}$, assumes the shape of a squared exponential kernel modified according to the smoothness and scale information obtained from $\mathcal{D}_{obs}$. First, define $k_a(x, x')$ as

$$
k_a(x, x') \equiv \lambda_a \times v_x \times v_{x'} \times \exp\left( -\frac{1}{2} \frac{(\hat{x} - \hat{x}')^2 + (\hat{y}_x - \hat{y}_{x'})^2}{\sigma_a} \right) + \delta(x - x')10^{-5}, \tag{6}
$$

where $(\lambda_a, \sigma_h)$ are hyperparameters, $\delta(\cdot)$ is the delta function, $v_x$ is a rescaling of $K_{obs}(x, x)^{1/2}$, $\hat{x}$ is a rescaling of $\mathcal{X}$ to the $[0, 1]$ interval, $\hat{y}_x$ is a rescaling of $\mu_{obs}(x)$ to the $[0, 1]$ interval. More precisely,

$$
\hat{x} \equiv \frac{x - \min(\mathcal{X})}{\max(\mathcal{X}) - \min(\mathcal{X})}, \hat{y}_x \equiv \frac{\mu_{obs}(x) - \min(\mu_{obs}(\mathcal{X}))}{\max(\mu_{obs}(\mathcal{X})) - \min(\mu_{obs}(\mathcal{X}))}, v_x = \sqrt{\frac{K_{obs}(x, x)}{\max_{x'} K_{obs}(x', x')}}.
\tag{7}
$$

Equation (6) is designed to borrow information from the (estimated) smoothness of $f(\mathcal{X})$, by decreasing the correlation of the distortion factors $a(x)$ and $a(x')$ as a function of the Euclidean distance between the 2D points $(x, \mu_{obs}(x))$ and $(x', \mu_{obs}(x'))$, properly scaled. Hyperparameter $\sigma_a$ controls how this distance is weighted. (6) also captures information about the amplitude of the distortion signal, making it proportional to the ratios of the diagonal entries of $K_{obs}(\mathcal{X})$. Hyperparameter $\lambda_a$ controls how this amplitude is globally adjusted. Nugget $10^{-5}$ brings stability to the sampling of $a(\mathcal{X})$ within Markov chain Monte Carlo (MCMC) inference. Hyper-hyperpriors on $\lambda_a$ and $\sigma_a$ are set as

$$
\log(\lambda_a) \sim \mathcal{N}(0, 0.5), \qquad \log(\sigma_a) \sim \mathcal{N}(0, 0.1). \tag{8}
$$

That is, $\lambda_a$ follows a log-Normal distribution with median 1, approximately 90% of the mass below 2.5, and a long tail to the right. The implied distribution for $a(x)$ where $s_x = 1$ will have most of its mass within a factor of 10 from its median. The prior on $\sigma_a$ follows a similar shape, but with a narrower allocation of mass. Covariance matrix $K_b$ is defined in the same way, with its own hyperparameters $\lambda_b$ and $\sigma_b$. Finally, the usual Jeffrey's prior for error variances is given to $\sigma_{int}^2$.

Figure 2 shows an example of inference obtained from synthetic data, generated according to the protocol of Section 4. In this example, the observational relationship between $X$ and $Y$ has the opposite association of the true causal one, but after adjusting for 15 of the 25 confounders that

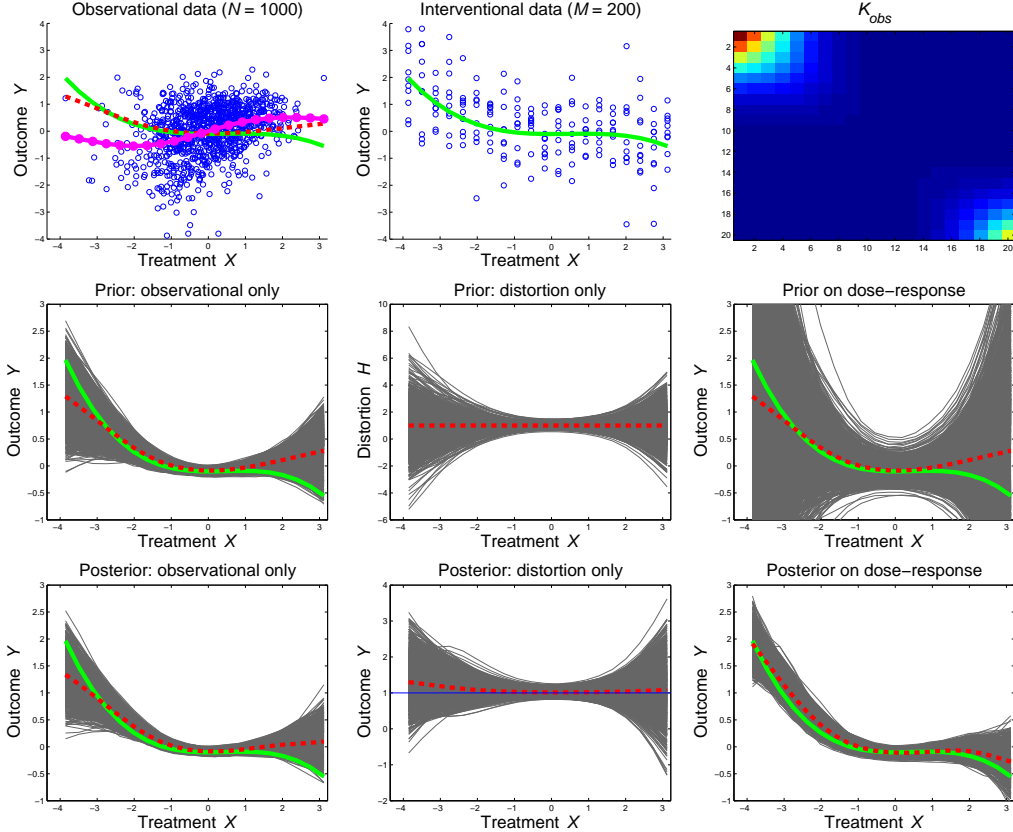

Figure 2: An example with synthetic data ($|\mathbf{Z}| = 25$), from priors to posteriors. Figure best seen in color. Top row: scatterplot of observational data, with true dose-response function in solid green, adjusted $\mu_{obs}$ in dashed red, and the unadjusted Gaussian process regression of $Y$ on $X$ in dashed-and-circle magenta (which is a very badly biased estimate in this example); scatterplot in the middle shows interventional data, 20 dose levels uniformly spread in the support of the observational data and 10 outputs per level − notice that the sign of the association is the opposite of the observational regime; matrix $K_{obs}$ is depicted at the end, where the nonstationarity of the process is evident. Middle row: priors constructed on $f_{obs}(\mathcal{X})$ and $a(\mathcal{X})$ with respective means; plot at the end corresponds to the implied prior on $a \odot f_{obs} + b$. Bottom row: the respective posteriors obtained by Gibbs sampling.

generated the data (10 confounders are randomly ignored to mimic imperfect prior knowledge), a reasonable initial estimate for $f(\mathcal{X})$ is obtained. The combination with interventional data results in a much better fit, but imperfections still exist at the strongest levels of treatment: the green curve drops at $x > 2$ stronger than the expected posterior mean. This is due to having both a prior derived from observational data that got the wrong direction of the dose-response curve at $x > 1.5$, and being unlucky at drawing several higher than expected values in the interventional regime for $x = 3$. The model then shows its strength on capturing much of the structure of the true dose-response curve even under misspecified adjustments, but the example provides a warning that only so much can be done given unlucky draws from a small interventional dataset.

### 3.3 Inference, Stratified Learning and Active Learning

In our experiments, we infer posterior distributions by Gibbs sampling, alternating the sampling of latent variables $f(\mathcal{X})$, $a(\mathcal{X})$, $b(\mathcal{X})$ and hyperparameters $\lambda_a, \sigma_a, \lambda_b, \sigma_b, \sigma_{int}^2$, using slice sampling [15] for the hyperparameters. The meaning of the individual posterior distribution over $f_{obs}(\mathcal{X})$ might also be of interest. This quantity is potentially identifiable by considering a joint model for $(\mathcal{D}_{obs}, \mathcal{D}_{int})$: in this case, $f_{obs}(\mathcal{X})$ learns the observational adjustment $\int g(x, \mathbf{z}) p(\mathbf{z})\, d\mathbf{z}$. This suggests that the posterior distribution for $f_{obs}(\mathcal{X})$ will change little according to model (5), which

is indeed observed in practice and illustrated by Figure 2. Learning the hyperparameters for $K_{obs}$ could be done jointly with the remaining hyperparameters, but the cost per iteration would be high due to the update of $K_{obs}$. The MCMC procedure for (5) is relatively inexpensive assuming that $|\mathcal{X}|$ is small. Learning the hyperparameters of $K_{obs}$ separately is a type of "modularization" of Bayesian inference [10].

As we mentioned in Section 2, it is sometimes desirable to learn dose-response curves conditioned on a few covariates $\mathbf{S} \subset \mathbf{Z}$ of interest. In particular, in this paper we will consider the case of straightforward stratification: given a set $\mathbf{S}$ of discrete covariates assuming instantiations $\mathbf{s}$, we have functions $f^{\mathbf{s}}(\mathcal{X})$ to be learned. Different estimation techniques can be used to borrow statistical strength across levels of $\mathbf{S}$, both for $f^{\mathbf{s}}(\mathcal{X})$ and $f_{obs}^{\mathbf{s}}(\mathcal{X})$. However, in our implementation, where we assume $|\mathbf{S}|$ is very small (a realistic case for many experimental designs), we construct independent priors for the different $f_{obs}^{\mathbf{s}}(\mathcal{X})$ with independent affine transformations.

Finally, in the Supplementary Material we also consider simple active learning schemes [11], as suggested by the fact that prior information already provides different estimates of uncertainty across $\mathcal{X}$ (Figure 2), which is sometimes dramatically nonstationary.

## 4   Experiments

Assessing causal inference algorithms requires fitting and predicting data generated by expensive randomized trials. Since this is typically unavailable, we will use simulated data where the truth is known. We divide our experiments in two types: first, one where we generate random dose-response functions, which allows us to control the difficulty of the problem in different directions; second, one where we start from a real world dataset and generate "realistic" dose-response curves from which simulated data can be given as input to the method.

### 4.1   Synthetic Data Studies

We generate studies where the observational sample has $N = 1000$ data points and $|\mathbf{Z}| = 25$ confounders. Interventional data is generated at three different levels of sample size, $M = 40$, 100 and 200 where the intervention space $\mathcal{X}$ is evenly distributed within the range shown by the observational data, with $|\mathcal{X}| = 20$. Covariates $\mathbf{Z}$ are generated from a zero-mean, unit variance Gaussian with correlation of $0.5$ for all pairs. Treatment $X$ is generated by first sampling a function $f_i(z_i)$ for every covariate from a Gaussian process, summing over $1 \leq i \leq 25$ and adding Gaussian noise. Outcome $Y$ is generated by first sampling linear coefficients and one intercept to weight the contribution of confounders $\mathbf{Z}$, and then passing the linear combination through a quadratic function. The dose-response function of X on Y is generated as a polynomial, which is added to the contribution of $\mathbf{Z}$ and a Gaussian error. In this way, it is easy to obtain the dose-response function analytically.

Besides varying $M$, we vary the setup in three other aspects: first, the dose-response is either a quadratic or cubic polynomial; second, the contribution of $X$ is scaled to have its minimum and maximum value spam either $50\%$ or $80\%$ of the range of all other causes of $Y$, including the Gaussian noise (a spam of $50\%$ already generates functions of modest impact to the total variability of $Y$); third, the actual data given to the algorithm contains only 15 of the 25 confounders. We either discard 10 confounders uniformly at random (the RANDOM setup), or remove the "top 10 strongest" confounders, as measured by how little confounding remains after adjusting for that single covariate alone (the ADVERSARIAL setup). In the interest of space, we provide a fully detailed description of the experimental setup in the Supplementary Material. Code is also provided to regenerate our data and re-run all of these experiments.

Evaluation is done in two ways. First, by the normalized absolute difference between an estimate $\hat{f}(x)$ and the true $f(x)$, averaged over $\mathcal{X}$. The normalization is done by dividing the difference by the gap between the maximum and minimum true values of $f(\mathcal{X})$ within each simulated problem[1]. The second measure is the log density of each true $f(x)$, averaged over $x \in \mathcal{X}$, according to the inferred posterior distribution approximated as a Gaussian distribution, with mean and variance estimated by MCMC. We compare our method against: I. a variation of it where $a$ and $b$ are fixed at $\mathbf{1}$ and $\mathbf{0}$, so the only randomness is in $f_{obs}$; II. instead of an affine transformation, we set $f(\mathcal{X}) = f_{obs}(\mathcal{X}) + r(\mathcal{X})$,

Table 1: For each experiment, we have either quadratic (Q) or cubic (C) ground truth, with a signal range of 50% or 80%, and an interventional sample size of $M = 40$, 100 and 200. $\mathbb{E}_i$ denotes the difference between competitor $i$ and our method regarding mean error, see text for a description of competitors. The mean absolute error for our method is approximately 0.20 for $M = 40$ and 0.10 for $M = 200$ across scenarios. $\mathcal{L}_i$ denotes the difference between our method and competitor $i$ regarding log-likelihood (differences > 10 are ignored, see text). That is, positive values indicate our method is better according to the corresponding criterion. All results are averages over 50 independent simulations, italics indicate statistically significant differences by a two-tailed t-test at level $\alpha = 0.05$.

| | Q50% RANDOM | | | Q50% ADV | | | Q80% RANDOM | | | Q80% ADV | | |
|---|---|---|---|---|---|---|---|---|---|---|---|---|
| | 40 | 100 | 200 | 40 | 100 | 200 | 40 | 100 | 200 | 40 | 100 | 200 |
| $\mathbb{E}_I$ | 0.00 | *0.02* | *0.01* | *0.07* | *0.07* | *0.05* | 0.00 | 0.00 | 0.01 | *0.05* | *0.04* | *0.03* |
| $\mathbb{E}_{II}$ | *0.05* | *0.02* | 0.01 | *0.04* | 0.00 | 0.00 | *0.04* | *0.03* | *0.02* | *0.04* | *0.02* | 0.00 |
| $\mathbb{E}_{III}$ | *0.11* | *0.07* | *0.03* | *0.05* | 0.01 | 0.01 | *0.11* | *0.06* | *0.03* | *0.08* | *0.03* | 0.01 |
| $\mathcal{L}_I$ | 2.33 | *2.31* | *2.18* | *7.16* | *6.68* | *6.23* | 0.62 | 0.53 | 0.45 | *2.16* | *1.79* | *1.50* |
| $\mathcal{L}_{II}$ | 0.78 | *0.28* | *0.17* | *0.44* | -0.17 | -0.16 | 0.53 | *0.42* | 0.20 | *0.25* | 0.07 | -0.09 |
| $\mathcal{L}_{III}$ | > 10 | > 10 | *0.43* | > 10 | > 10 | -0.06 | 0.74 | *0.44* | *0.36* | *0.33* | -0.01 | -0.10 |

| | C50% RANDOM | | | C50% ADV | | | C80% RANDOM | | | C80% ADV | | |
|---|---|---|---|---|---|---|---|---|---|---|---|---|
| | 40 | 100 | 200 | 40 | 100 | 200 | 40 | 100 | 200 | 40 | 100 | 200 |
| $\mathbb{E}_I$ | 0.01 | *0.02* | *0.03* | *0.08* | *0.08* | *0.07* | *0.03* | *0.05* | *0.05* | *0.09* | *0.09* | *0.08* |
| $\mathbb{E}_{II}$ | *0.05* | *0.03* | *0.02* | *0.05* | *0.02* | 0.01 | *0.05* | *0.03* | *0.02* | *0.07* | *0.03* | *0.02* |
| $\mathbb{E}_{III}$ | *0.08* | *0.04* | *0.04* | 0.03 | *0.04* | *0.02* | *0.11* | *0.06* | *0.03* | *0.09* | *0.05* | *0.02* |
| $\mathcal{L}_I$ | > 10 | > 10 | > 10 | *9.62* | *9.05* | *8.68* | > 10 | > 10 | > 10 | > 10 | > 10 | > 10 |
| $\mathcal{L}_{II}$ | *3.49* | *0.83* | *0.41* | *4.45* | *0.43* | -0.10 | *1.07* | *0.64* | -0.04 | *0.96* | *0.30* | *0.14* |
| $\mathcal{L}_{III}$ | > 10 | > 10 | > 10 | > 10 | > 10 | > 10 | > 10 | *0.79* | 0.03 | *0.45* | *0.18* | -0.03 |

where $r$ is given a generic squared exponential Gaussian process prior, which is fit by marginal maximum likelihood; III. Gaussian process regression with squared exponential kernel applied to the interventional data only and hyperparameters fitted by marginal likelihood. The idea is that competitors I and II provide sensitivity analysis of whether our more specialized prior is adding value. In particular, competitor II would be closer to the traditional priors used in computer-aided experimental design [2] (but for our specialized $K_{obs}$). Results are shown in Table 1, according to the two assessment criteria, using $\mathbb{E}$ for average absolute error, and $\mathcal{L}$ for average log-likelihood.

Our method demonstrated robustness to varying degrees of unmeasured confounding. Compared to Competitor I, the mean obtained without any further affine transformation already provides a competitive estimator of $f(\mathcal{X})$, but this suffers when unmeasured confounding is stronger (ADVERSARIAL setup). Moreover, uncertainty estimates given by Competitor I tend to be overconfident. Competitor II does not make use of our special covariance function for the correction, and tends to be particularly weak against our method in lower interventional sample sizes. In the same line, our advantage over Competitor III starts stronger at $M = 40$ and diminishes as expected when $M$ increases. Competitor III is particularly bad at lower signal-to-noise ratio problems, where sometimes it is overly confident that $f(\mathcal{X})$ is zero everywhere (hence, we ignore large likelihood discrepancies in our evaluation). This suggests that in order to learn specialized curves for particular subpopulations, where $M$ will invariably be small, an end-to-end model for observational and interventional data might be essential.

## 4.2 Case Study

We consider an adaptation of the study analyzed by [7]. Targeted at premature infants with low birth weight, the Infant Health and Development Program (IHDP) was a study of the efficacy of "educational and family support services and pediatric follow-up offered during the first 3 years of life" [3]. The study originally randomized infants into those that received treatment and those that did not. The outcome variable was an IQ test applied when infants reached 3 years. Within those which received treatment, there was a range of *number of days* of treatment. That dose level was not randomized, and again we do not have ground truth for the dose-response curve. For our assessment, we fit a dose-response curve using Gaussian processes with Gaussian likelihood function and the back-door adjustment (3) on available covariates. We then use the model to generate independent synthetic "interventional data." Measured covariates include birth weight, sex, whether the mother smoked during pregnancy, among other factors detailed by [7, 3]. The Supplementary Material goes in detail about the preprocessing, including R/MATLAB scripts to generate the data. The

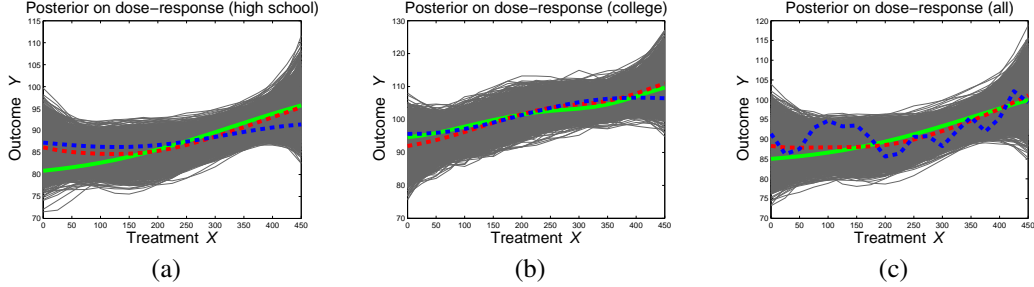

Figure 3: An illustration of a problem generated from a model fitted to real data. That is, we generated data from "interventions" simulated from a model that was fitted to an actual study on premature infant development [3], where the dose is the number of days that an infant is assigned to follow a development program and the outcome is an IQ test at age 3. (a) Posterior distribution for the stratum of infants whose mothers had up to some high school education, but no college. The red curve is the posterior mean of our method, and the blue curve the result of Gaussian process fit with interventional data only. (b) Posterior distributions for the infants whose mothers had (some) college education. (c) The combined strata.

observational sample contained 347 individuals (corresponding only to those which were eligible for treatment and had no missing outcome variable) and 21 covariates. This sample included 243 infants whose mother attended (some) high school but not college, and 104 with at least some college.

We generated 100 synthetic interventional datasets stratified by mother's education, (some) high-school vs. (some) college. 19 treatment levels were pre-selected (0 to 450 days, increments of 25 days). All variables were standardized to zero mean and unit standard deviation according to the observational distribution per stratum. Two representative simulated studies are shown in Figure 3, depicting dose-response curves which have modest evidence of non-linearity, and differ in range per stratum[2]. On average, our method improved over the fitting of a Gaussian process with squared exponential covariance function that was given interventional data only. According to the average normalized absolute differences, the improvement was $0.06$, $0.07$ and $0.08$ for the high school, college and combined data, respectively (where error was reduced in $82\%$, $89\%$ and $91\%$ of the runs, respectively), each in which 10 interventional samples were simulated per treatment level per stratum.

## 5  Conclusion

We introduced a simple, principled way of combining observational and interventional measurements and assessed its accuracy and robustness. In particular, we emphasized robustness to model misspecification and we performed sensitivity analysis to assess the importance of each individual component of our prior, contrasted to off-the-shelf solutions that can be found in related domains [2].

We are aware that many practical problems remain. For instance, we have not discussed at all the important issue of sample selection bias, where volunteers for an interventional study might not come from the same $p(\mathbf{Z})$ distribution as in the observational study. Worse, neither the observational nor the interventional data might come from the population in which we want to enforce a policy learned from the combined data. While these essential issues were ignored, our method can in principle be combined with ways of assessing and correcting for sample selection bias [1]. Moreover, if unmeasured confounding is too strong, one cannot expect to do well. Methods for sensitivity analysis of confounding assumptions [13] can be integrated with our framework. A more thorough analysis of active learning using our approach, particularly in the light of possible model misspecification, is needed as our results in the Supplementary Material only superficially covers this aspect.

**Acknowledgments**

Thanks to Jennifer Hill for clarifications about the IHDP data, and Robert Gramacy for several useful discussions.

## Footnotes

[1]Data is also normalized to a zero mean, unit variance according to the empirical mean and variance of the observational data, in order to reduce variability across studies.

[2]We do *not* claim that these curves represent the true dose-response curves: confounders are very likely to exist, as the dose level was not decided at the beginning of the trial and is likely to have been changed "on the fly" as the infant responded. It is plausible that our covariates cannot reliably account for this feedback effect.

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
