[Supplementary Material]

# Supplementary Material of "Observational-Interventional Priors for Dose-Response Learning"

**Ricardo Silva**

Department of Statistical Science and Centre for Computational Statistics and Machine Learning
University College London
ricardo@stats.ucl.ac.uk

## Abstract

In this Supplementary Material, we discuss: i. a detailed explanation of our synthetic data generation protocol; ii. a detailed explanation of our preprocessing of the Infant Health and Development Program dataset; iii. an illustration of active learning using our approach; iv. an illustrative comparison of our method against existing methods for deep Gaussian processes in the literature; v. details of the Gibbs sampler.

## 1 Synthetic Data Generator

We generate data from a multivariate distribution where $X$ is the treatment, $Y$ is the outcome, and $\mathbf{Z}$ are covariates that cause $X$ and $Y$. The model for the covariates is

$$\mathbf{Z} \sim \mathcal{N}(\mathbf{0}, \Sigma_{\mathbf{Z}}),$$

where $\Sigma_{\mathbf{Z}}$ is a correlation matrix with every off-diagonal entry equal to $0.5$.

The model for $\mathbf{X}$ given $\mathbf{Z}$ is

$$X = \sum_{i=1}^{p} f_{x_i}(z_i) + e_X,$$

where $p = |\mathbf{Z}|$ and $e_X \sim N(0, \sigma_x^2)$. Each function $f_{x_i}(\cdot)$ is first sampled at the realized values of $Z_i$ from a zero-mean Gaussian process prior with covariance function $k(z_i, z_i') \equiv \exp(-(z_i - z_i')^2/4)$, then divided by $\sqrt{p}$ so that the variance of the function generation process does not grow with $p$. We then calculate the empirical variance $v_{f_x}$ of $\sum_i f_{x_i}(Z_i)$ in the sample generated, and set $\sigma_x^2 = b \times v_{f_x}$, where $b \sim \mathcal{U}(0.2, 0.4)$, the uniform distribution in the interval $[0.2, 0.4]$. In this way, causes of $X$ that are not causes of $Y$ (that is, $e_X$) contribute to the variance of $X$ with approximately 20% to 40% of the variance contributed by the common causes.

The next step is to generate

$$\theta_i \sim \mathcal{N}\left(0, \frac{1}{p+1}\right),$$

for $0 \le i \le p$, and

$$\beta_i \propto \mathcal{N}(0,1)I(|\beta_i| > 0.2),$$

$i \in \{0,1,2\}$ and $I(\cdot)$ the indicator function. That is, each $\beta_i$ comes from a standard Gaussian restricted to the space $|\beta_i| > 0.2$. We then define

$$
\begin{aligned}
Z_y &\equiv \theta_0 + \theta_{1:p}^\top \mathbf{Z} \\
f_{yz} &\equiv \beta_2 Z_y^2 + \beta_1 Z_y + \beta_0 \\
f_{yze} &\equiv f_{yz} + e_Y \\
e_Y &\sim \mathcal{N}(0, \sigma_y^2).
\end{aligned}
$$

Quantity $f_{yze}$ is the contribution of "all other causes" of $Y$ but $X$. Analogously to $\sigma_x^2$, we set $\sigma_y^2 = b' \times v_{f_{yz}}$, where $b' \sim \mathcal{U}(0.2, 0.4)$ and $v_{f_{yz}}$ is the empirical variance of the sampled values of $f_{yz}$. What is left is the contribution of $X$ according to

$$Y = f_{yx}(X) + f_{yze},$$

in a way we can control (up to some point) how much $X$ contributes to the variability of $Y$. Function $f_{yx}(\cdot)$ is set to be a polynomial of degree $d$. In our experiments, we set $d = 2$ and $d = 3$.

Let $\alpha$ be a number between 0 and 0.5. Let $R_\alpha$ and $R_{1-\alpha}$ be the corresponding empirical quantiles of $f_{yze}$. Define $R \equiv R_{1-\alpha} - R_\alpha$. In our experiments, we choose either $\alpha = 0.1$ or $\alpha = 0.25$. We constraint our $f_{yze}$ to be within a range of length $R$ as follows. For any realization $x$ of $X$, define $\hat{x}$ as the standardization of $x$ according to the empirical mean and variance of the sampled values of $X$. That is, given the empirical mean $\hat{m}$ of the sampled values of $X$ and the empirical variance $\hat{v}$, $\hat{x} \equiv (x - \hat{m})/\sqrt{\hat{v}}$. Both $\hat{m}$ and $\hat{v}$ become extra parameters of $f_{yx}(\cdot)$. Given a degree $d$, we set

$$
\begin{aligned}
\lambda_i' &\propto \mathcal{N}(0, 1)I(|\lambda_i| > 0.2) \\
f_{yx}'(x) &\equiv \sum_{i=0}^{d} \lambda_i' \hat{x}^i, \\
R' &\equiv \max f_{yx}'(\hat{x}) - \min f_{yx}'(\hat{x}) \\
\lambda_i &\equiv \alpha_i' \times \frac{R}{R'} \\
f_{yx}(x) &\equiv \sum_{i=0}^{d} \lambda_i \hat{x}^i.
\end{aligned}
$$

In the third line of the above, the maximum and minimum operations are taken over the empirical samples of $X$. The end result is a function that first linearly transforms $X$ to a more standard scale and location, then passes it to a polynomial function with a range is approximately of the same length as the difference between the $1 - \alpha$ and $\alpha$ quantiles of the realizations of $f_{yze}$. Setting $\alpha$ to values close to 0.5 would make the signal due to $X$ to be mostly constant, its variability almost undetectable compared to the variability of the other causes of $Y$. Finally, we reject this model and redo the model generating process if the absolute value of the empirical rank correlation between the samples of $X$ and $f_{yz}$ is less than 0.2, so that a minimal degree of confounding is enforced.

Notice that the motivation for setting $\mathbf{Z} \sim \mathcal{N}(\mathbf{0}, \Sigma_{\mathbf{Z}})$, and $f_{yz}(\mathbf{Z})$ to a quadratic function, is to allow us to analytically calculate $\mathbb{E}[f_{yz}(\mathbf{Z})]$. This is important, since

$$\mathbb{E}[Y \mid do(X = x)] = f_{yx}(x) + \mathbb{E}[f_{yz}(\mathbf{Z})],$$

the value of which is necessary for a precise calculation of the estimation error.

As part of the Supplementary Material, we provide MATLAB code to reconstruct the experiments. This is done via the function `generate_problems.m`, which can also make use of a file that provides the seed to reconstruct the synthetic models and data exactly.

To complement the results in the main text, Table 1 shows further comparisons. Method IV is the one obtained by just fitting the observational data for treatment and outcome, assuming no confounding (that is, no back-door adjustment is done). It provides a sense of the difficulty of the generated problems. Method V is yet another sensitivity analysis, now for the role of $a$. This is done by effectively dropping $a$ from the mapping between $f_{obs}$ and $f$ (that is, the generation of $f$ is defined as $f(\mathcal{X}) \equiv f_{obs}(\mathcal{X}) + b(\mathcal{X})$). It differs from Method III in the main text by giving $b$ a non-stationary covariance function derived from $\mathcal{D}_{obs}$, as opposed to the off-the-shelf squared exponential used by Method III. It is clear that although the $a$ component does not seem to help (or hurt) the decrease of the absolute error, it makes a significant difference in terms of modeling the posterior uncertainty. Differences are more prominent under the ADVERSARIAL regime, which can be partially explained by the heavy-tailed, non-Gaussian nature of the product $a \odot f_{obs}$. We emphasize that our measure $\mathcal{L}_V$ is *per point* $x \in \mathcal{X}$, and that even a difference of 0.05 in average log-likelihood means a ratio of densities of 2.7 in the original scale, for $|\mathcal{X}| = 20$, and a ratio of approximately $20,000$ for a difference of 0.50.

Table 1: A table analogous to the one found in the main text, Section 4. Here, method IV is just the dose-response obtained by fitting the observational data only without any back-door adjustment. Method V is the method where we set $a \equiv 1$, inferring $b$ only.

| | Q50% RANDOM | | | Q50% ADV | | | Q80% RANDOM | | | Q80% ADV | | |
|---|---|---|---|---|---|---|---|---|---|---|---|---|
| | 40 | 100 | 200 | 40 | 100 | 200 | 40 | 100 | 200 | 40 | 100 | 200 |
| $\mathbb{E}_{IV}$ | 0.41 | 0.45 | 0.47 | 0.36 | 0.41 | 0.44 | 0.30 | 0.34 | 0.37 | 0.28 | 0.33 | 0.36 |
| $\mathbb{E}_{V}$ | -0.01 | 0.00 | 0.00 | 0.01 | 0.01 | 0.01 | 0.00 | 0.00 | 0.00 | 0.00 | 0.01 | 0.01 |
| $\mathcal{L}_{V}$ | -0.01 | 0.04 | 0.08 | 0.31 | 0.47 | 0.55 | 0.01 | 0.07 | 0.08 | 0.21 | 0.32 | 0.37 |

| | C50% RANDOM | | | C50% ADV | | | C80% RANDOM | | | C80% ADV | | |
|---|---|---|---|---|---|---|---|---|---|---|---|---|
| | 40 | 100 | 200 | 40 | 100 | 200 | 40 | 100 | 200 | 40 | 100 | 200 |
| $\mathbb{E}_{IV}$ | 0.30 | 0.32 | 0.35 | 0.27 | 0.30 | 0.33 | 0.30 | 0.34 | 0.37 | 0.28 | 0.33 | 0.36 |
| $\mathbb{E}_{V}$ | 0.00 | 0.00 | 0.00 | 0.01 | 0.00 | 0.00 | 0.00 | 0.00 | 0.01 | 0.00 | 0.01 | 0.01 |
| $\mathcal{L}_{V}$ | -0.02 | 0.01 | 0.04 | 0.19 | 0.25 | 0.36 | 0.06 | 0.14 | 0.30 | 0.22 | 0.40 | 0.50 |

Figure 1: Example of synthetic data sampled from the three models (stratified by mother's education, and then combined). The amount of variability around each response is similar to the one found around the observational regression curve. Curve represents the synthetic dose-response curve fitted to each scenario based on an observational sample of size $347$.

## 2 Preprocessing of the Infant Health and Development Program Data

The original Infant Health and Development Program (IHDP) data can be downloaded from `http://www.icpsr.umich.edu/icpsrweb/HMCA/studies/9795`. We start instead from the pre-processed version done by [3] and available[1] at `http://www.tandfonline.com/doi/suppl/10.1198/jcgs.2010.08162`. This data contains 985 individuals, or which 377 were given treatment. 30 individuals had missing outcome data. We discarded them to obtain a final sample size of 347. We applied further preprocessing to this data, to remove variables which we believed would be less relevant to our simulation (for instance, the home site of the family at the start of the intervention). Some variables were binarized, as we were concerned about the sample size. This includes some originally discrete, non-binary, variables, such as race. A detailed R script that loads the original file provided by [3] and performs the further processing is provided with this supplement (`process_ihdp.R`).

This resulted in a dataset with 21 columns. We fit a nonparametric model for the regression function $g(x, \mathbf{z})$ using a Gaussian process prior and Gaussian likelihood. The prior is the same as all other experiments, a Matérn $\frac{3}{2}$ covariance function with automatic relevance determination priors [5]. We fit all hyperparameters by marginal maximum likelihood using the GPML[2] package for MATLAB. The range of days of treatment in the treated IHDP subgroup varied from 0 to 468. We defined our set $\mathcal{X}$ of interventional levels at $0, 25, 50, \ldots, 450$.

To build a simulator for outcome variable $Y$, IQ score at age 3 (standardized by centering and scaling it according the the empirical mean and standard deviation of the observational data), we build a mean function $f(x)$ and error variance $\sigma_f^2$ from the fitted response function evaluated at the empirical

observational distribution,

$$f(x) \equiv \frac{1}{347} \sum_{i=1}^{347} \hat{g}(x, \mathbf{z}^{(i)}), x \in \mathcal{X}.$$

Less straightforward is deciding on a realistic choice of $\sigma_f^2$. First, it should be pointed out that as implied by the fitted observational model as ground truth,

$$Y \mid do(x), \mathbf{z} \sim \mathcal{N}(\hat{g}(x, \mathbf{z}), \hat{\sigma}_Y^2),$$

where $\hat{\sigma}_Y^2$ is given by GPML, that $Y \mid do(x)$ will in general have heteroscedastic variance (if $\hat{g}(x, \mathbf{z})$ is not additive in $X$), or even be non-Gaussian distributed. To deal with that, we calculate the empirical variance of $\{\hat{g}(x, \mathbf{z}^{(1)}), \dots, \hat{g}(x, \mathbf{z}^{(347)})\}$ for each $x \in \mathcal{X}$, and set $\sigma_f^2$ to be the average of these quantities plus the error variance of the regression of $Y$ on $X$ and $\mathbf{Z}$. Normality is used as a convenient approximation for the resulting model $Y \mid do(x)$. Heteroscedastic regression can be adopted by our framework without any conceptual changes, but we ignore it for convenience of presentation.

# 3 Active Learning Illustration

The probabilistic formulation of our dose-response model leads to Bayesian active learning schemes where observational data $\mathcal{D}_{obs}$ is fixed and new measurements are continuously added to interventional dataset $\mathcal{D}_{int}$. In this Section, we provide an illustration on how to use our model with the simplest design scheme: the "D-optimal" design where the next dose level $x$ to be picked is the one corresponding to target $f(x)$ of highest entropy. A classical review of the motivations and shortcomings of several designs from a Bayesian perspective is given by [4].

To approximate the entropy of a given $f(x)$, we merely compute its estimated variance from the current MCMC samples as we observe that in the posterior the marginal distributions of each $f(x)$ are not too dissimilar from Gaussians, or at least can be ranked based on variances alone. Use of the variance can be formally justified by standard second-order approximations [4] even if we still rely on MCMC samples.

We applied this idea to our IHDP problem, where we initialize the model by sampling one outcome for each dose level $x \in \mathcal{X}$. We then are given a budget of $5 \times |\mathcal{X}| = 95$ trials to spend. For every new dose level selected, we "run the intervention" using our simulated model, and collect a new data point. We update the distribution of the latent variables at every new point collected, but to save time we update the distribution of the hyperparameters only after 5 new points have been collected. The budget of 95 points is shared across the two strata. In our provided MATLAB code, function `dose_response_learning_stratified.m` implements this scheme.

In Figure 2, we show how treatments were allocated to each stratum, and how they were distributed. As expected, most of the doses were given at the endpoints of $\mathcal{X}$. Stratum "high school" was allocated 31 of the 95 (simulated) trials, with the remaining 64 given to the "college" stratum. We compare it against the policy of allocating an equal number (6) of trials to each of the 19 levels of $\mathcal{X}$. Figure 3 illustrates the posterior distributions for the samplers given one actively selected set and one uniformly selected set. While the differences are not major, it is clear that the active scheme does better or at least as well even in regions were no more than two datapoints have been collected, with a clear advantage in regions where the prior was not able to capture the true curve (lower levels of stratum "high school").

# 4 A Note on Generic Deep Gaussian Processes

The transformation given by $a$ and $b$ is not identifiable: like a deep Gaussian process prior [2], its usefulness comes from providing an adequate prior distribution for $f$ that we evaluated at length through a series of comparisons and sensitivity analyzes.

In any case, this raises the question of directly adopting the generic transformation of $f_{obs}(\mathcal{X})$,

$$f(x) = u(f_{obs}(x)), x \in \mathcal{X},$$

where $u(\mathcal{X})$ is a function that is given a Gaussian process prior. One appropriate choice of mean for this process is the identity function, $\mu_u(f_{obs}(x)) = f_{obs}(x)$, with the covariance matrix $K_u$

Figure 2: Histogram of the allocation of 114 experiments (initial 19 followed by adaptively selected 95 further trials) in two different conditions according to our simple active learning criteria.

Figure 3: Corresponding models learned from this data. The red curve corresponds to the expected dose-response according to the collected sample, while the blue curve is the result of our procedure with a given set of 133 uniformly sampled at our $\mathcal{X}$ grid of 19 dose levels. The top row illustrates samples from the posterior learned from the active selection, the bottom row are samples from the posterior learned from the uniform selection. In general, there is a slight advantage for the active selection at this sample size, as the posterior typically allocates higher probability to the true curve.

constructed from smooth covariance functions, as we want to bias this prior toward the (unknown) observational curve $f_{obs}(\mathcal{X})$.

It is not clear, however, why this generic construction would have advantages over our pointwise affine prior. The original motivation for deep learning is to combine signals from a high-dimensional space, and here our treatment is a scalar dosage. Our goal in this section is just to provide a simple illustration that, for a dose-response curve where the signal is just a scalar, there is no obvious reason to use more complicated models.

Sampling $f_{obs}(\cdot)$ in this "deep" setup is difficult due to its appearance on $K_u$. We illustrate the advantages of our pointwise affine prior with a simple experiment, once again based on the IHDP data. We define $K_u$ with a squared exponential covariance function,

$$k_u(f_{obs}(x), f_{obs}(x')) \equiv \lambda_u \times \exp\left(-\frac{1}{2}\frac{(f_{obs}(x) - f_{obs}(x'))^2}{\sigma_u}\right) + \delta(f_{obs}(x) - f_{obs}(x'))10^{-5}$$

Figure 4: A comparison of results for the IHDP data using a more standard deep Gaussian process prior against our affine transformation prior. For the more standard deep CP, Hamiltonian MCMC was used. Each plot show 200 sampled dose-response curves. In the affine case, these correspond to thinning a run of 2000 iterations by skipping 10 samples from every sample held.

with priors $\log(\lambda_u) \sim \mathcal{N}(0, 0.5)$ and $\log(\sigma_u) \sim \mathcal{N}(0, 0.1)$. Moreover, we rescale the covariance matrix of $f_{obs}(\mathcal{X})$ so that the largest entry of its diagonal is now 1. This is to give the standard deep GP an extra help, as exploring the posterior of $f_{obs}(\cdot)$ and the hyperparameters would be even harder with a more concentrated prior. We also enforce no parameter sharing of any kind among the different strata. In what follows, we do not claim that this prior is optimal for learning the dose-response curve, but as a convenient way of facilitating sampling for this model.

In Figure 4, we show posterior samples for the standard Gaussian process prior using the default Hamiltonian MCMC (HMC) methods implemented in Stan [1]. The dataset given contains 10 points per dose level of $\mathcal{X}$ in each of the three scenarios (190 per study, in total). Due to the high cost of performing sampling even in these modest datasets, we run HMC only for 220 iterations, discarding the first 20 iterations as burn-in. We run the off-the-shelf Gibbs with the slice sampling algorithm for our affine model. In Figure 4, we show the corresponding output obtained by running it for 2200 iterations, discarding the first 200, and then uniformly thinning the remaining 2000 iterations to obtain 200 samples.

It is clear that in Figure 4 that the affine prior performs substantially better. However, we do not want to make overgeneralized claims of inferential superiority, but to merely illustrate that we see no evidence that a standard deep Gaussian process prior would present any advantage. This is even more evident from the computational cost of both procedures. The HMC execution, even in the highly optimized Stan code, took approximately 1200 seconds in a 5-year old Xeon workstation, while inference with the factorized prior took two orders of magnitude less, 54 seconds. While powerful approximation algorithms can be applied to standard deep Gaussian processes [2], we recommend avoiding them, as in causal inference we are interested in parameter learning instead of merely predictive performance and the more precise calculation of credible intervals provided by MCMC is preferred to a variational approximation that will underestimate uncertainty.

# 5 MCMC Updates

In this Section, we present the updates for the MCMC algorithm from Section 3.3 of the main paper. Recall that the model is given by

$$
\begin{aligned}
f_{obs}(\mathcal{X}) &\sim \mathcal{N}(\mu_{obs}, K_{obs}) \\
a(\mathcal{X}) &\sim \mathcal{N}(\mathbf{1}, K_a) \\
b(\mathcal{X}) &\sim \mathcal{N}(\mathbf{0}, K_b) \\
f(\mathcal{X}) &= a(\mathcal{X}) \odot f_{obs}(\mathcal{X}) + b(\mathcal{X}) \\
Y_{int}^{(i)} &\sim \mathcal{N}(f(x_{int}^{(i)}), \sigma_{int}^2), 1 \le i \le M.
\end{aligned}
$$

As we take several measurements per position $x$, it is possible that $x^{(i)} = x^{(j)}$ for $i \ne j$. Given interventional dataset $\mathcal{D}_{int}$, let $\mathbf{Y}$ be the $M$-dimensional vector that collects all interventional outcomes, and let $A$ be the $M \times T$ binary matrix such that $A_{ik} = 1$ if and only if $x^{(i)} = x_k$, the $k$-th entry of $\mathcal{X}$. Finally, for convenience we will write vectors $f(\mathcal{X}), a(\mathcal{X})$ and $b(\mathcal{X})$ as $\mathbf{f}$, $\mathbf{a}$ and $\mathbf{b}$. That is,

$$
\mathbf{Y} \mid \{\mathbf{a}, \mathbf{b}, \mathbf{f}\} \sim \mathcal{N}(A(\mathbf{a} \odot \mathbf{f} + \mathbf{b}), \sigma_{int}^2 \mathbf{I}_M),
$$

where $\mathbf{I}_M$ is the $M \times M$ identity matrix. The conditional density of $\mathbf{f}$ given everything else is a Gaussian with covariance matrix $\Sigma_{\mathbf{f}}^{\mathcal{D}_{int}}$,

$$
(\Sigma_{\mathbf{f}}^{\mathcal{D}_{int}})^{-1} = K_{obs}^{-1} + \frac{\mathbf{I_a} A^\top A \mathbf{I_a}}{\sigma_{int}^2},
$$

where $\mathbf{I_a}$ is the diagonal matrix having $\mathbf{a}$ as the diagonal. The posterior mean $\mu_{\mathbf{f}}^{\mathcal{D}_{int}}$ is given by

$$
\mu_{\mathbf{f}}^{\mathcal{D}_{int}} = \Sigma_{\mathbf{f}}^{\mathcal{D}_{int}} \left( K_{obs}^{-1} \mu_{obs} + \frac{\mathbf{I_a} A^\top (\mathbf{Y} - A\mathbf{b})}{\sigma_{int}^2} \right).
$$

The posterior for $\mathbf{a}$ is also Gaussian with an analogous shape,

$$
(\Sigma_{\mathbf{a}}^{\mathcal{D}_{int}})^{-1} = K_a^{-1} + \frac{\mathbf{I_f} A^\top A \mathbf{I_f}}{\sigma_{int}^2},
$$

$$
\mu_{\mathbf{a}}^{\mathcal{D}_{int}} = \Sigma_{\mathbf{a}}^{\mathcal{D}_{int}} \left( K_a^{-1} \mathbf{1} + \frac{\mathbf{I_f} A^\top (\mathbf{Y} - A\mathbf{b})}{\sigma_{int}^2} \right).
$$

Finally, the posterior for $\mathbf{b}$ is also Gaussian with covariance matrix and mean given by

$$
(\Sigma_{\mathbf{b}}^{\mathcal{D}_{int}})^{-1} = K_b^{-1} + \frac{A^\top A}{\sigma_{int}^2},
$$

$$
\mu_{\mathbf{b}}^{\mathcal{D}_{int}} = \Sigma_{\mathbf{b}}^{\mathcal{D}_{int}} \left( \frac{A^\top (\mathbf{Y} - A(\mathbf{a} \odot \mathbf{f}))}{\sigma_{int}^2} \right).
$$

The posterior for $\sigma_{int}^2$ and the hyperparameters of $K_a$, $K_b$ has no special shape. We just apply slice sampling to each of these hyperparameters separately.

## Footnotes

[1]The corresponding file name in the supplement provided by Hill is `example.dat`, a R binary file.

[2]`http://www.gaussianprocess.org/gpml/code/matlab/doc/`