[Reviews · NeurIPS 2016]

Reviewer 1

Summary

The paper provides a principled way of fusing observational and experimental data so as to make a stronger claim about the effect of a treatment X on an outcome Y, P(y | do(X)). One key assumption is the existence of an admissible set Z for back-door adjustment, which means that observational data alone would be asymptotically sufficient to make about an experimental claim through the back-door formula. The authors noticed and showed, however, that in practice, it's still challenging to make a robust claim about P(y | do(X)) due to statistical variability. The authors then showed how to use easily available observational data to help leverage the scarcely available experimental data so as to make the best possible claim about the target do-distribution. The paper pursues a Bayesian approach and uses the observational dataset as a prior to be combined with the experimental dataset. The authors showed that the fusion of both datasets yields more robust and stable results than each individual estimation procedures alone.

Qualitative Assessment

The motivation for studying this problem is clear and real. The results are solid, and the presentation is smooth and neat. The authors seem to have a clear and ambitious big picture in mind. I also would be interested in learning more about the possible relaxations of the assumptions used in the paper, including but not limited to lack of confounding and sampling selection biases.

Confidence in this Review

2-Confident (read it all; understood it all reasonably well)


Reviewer 2

Summary

This paper proposes a two-layered prior model for dose-response learning. A Gaussian process is first learned from observational data and then is reshaped with a nonparametric affine transformation from controlled interventions. The key modelling idea is expressed in Eq. (5), where a nonstationary covariance function is constructed from the observational data. The advantages of the newly proposed model is empirically demonstrated by two examples involving simulation.

Qualitative Assessment

Generally speaking the paper is written pretty clearly (except some questions and minor issues that I will list at the end). The underlying reason of the proposed model can be easily understood and the two case studies (which both involve simulations) have shown some advantages of the proposed model. My main concern is that the idea of the paper does not seem significantly novel. The “elementwise affine” transform a . f_{obs} + b is simple and borrowing the information from K_{obs} to k_a also seems a natural way to address the problem. Overall this is more like developing a model suitable for one particular kind of problems. In addition, since both of the two examples involve simulation, the performance of the model has not been evaluated in any real data. (After all, the competitors to the proposed model in the paper seem naturally worse just from their semantics.) Some questions and minor issues that I have found in the paper are listed as follows. (1). In Line 110: Why do you choose a Matern 3/2 covariance function for g(.,.)? Have you tried other (common) covariance options or Matern with different parameters for the comparison before choosing a Matern 3/2 covariance? (2). In Line 120: What is the reason that “the cost of computing the covariance matrix K_{obs} of f_{obs}(X) is O(T^2 N^2)”? Do you assume that the TN * TN covariance matrix of the joint predictive distribution of {g(x, z^{(i)}} (mentioned at the end of Line 114) has already been computed? What is the running time of obtaining the posterior over functions g(.,.) (in your two studied cases)? (3). For Eq. (6) (which is right after Line 135) 10^{-5} seems a magic number. Should it depend on the learned value of \lambda_a? How do you choose such a number? (4). In Line 136: In “(\lambda_a, \sigma_h) are hyperparameters”, \sigma_h should be \sigma_a. (5). Right after Line 146: Again, 0.5 in N(0, 0.5) and 0.1 in N(0, 0.1) are magic numbers. Should they be fixed or adjusted according to different data cases? (6). In the third line of the caption for Figure 2: What is the meaning of “adjusted” in “adjusted \mu_{obs}”? What is the meaning of “unadjusted” in “unadjusted Gaussian process regression”?

Confidence in this Review

2-Confident (read it all; understood it all reasonably well)


Reviewer 3

Summary

In this paper, the authors propose to use Gaussian process models to estimate a continuous counterfactual treatment-outcome relationship from a combination of interventional and observational data.

Qualitative Assessment

The paper is very well-written, clear, and has good experimental validation. All the code is provided in the supplement. I have little to add, I think this is a very strong paper, and the authors clearly put in a lot of effort and care into writing it. I think the natural question raised by the author's paper is how to bring in ideas from the design of experiments community (which often uses Gaussian process models for complex treatment outcome relationships, but generally does not use observational data or causal models that allow them to exploit this data) to make active learning by interventions better.

Confidence in this Review

2-Confident (read it all; understood it all reasonably well)


Reviewer 4

Summary

In this paper, for dose-response learning, the authors proposed a hierarchical Gaussian process prior that constructs a distribution over the dose-response curve by learning from observational data, and reshapes the distribution with a nonparametric affine transform learned from controlled interventions.

Qualitative Assessment

The experiments is not convincing. First, the authors only compared their methods against its variations, did not compare with other methods in the related work, such as [1],[2]. Second, the experimental results in Table 1 only shown the difference between baselines and proposed method regarding mean error or log-likelihood, we do not know the real mean error of the proposed method, which make it hard to know the performance of the proposed method. Third, the sample size of experiments is up to 1000 in the synthetic data studies. And there are many practical problems remain about the proposed method. First, it is hardly to collect the interventional data where the control variable has many different levels. Second, the observational data and interventional data might come form different population, which make the difference distribution of p(Z) between these two datasets. [1]. Ernest J, Bühlmann P. Marginal integration for nonparametric causal inference[J]. Electronic Journal of Statistics, 2015, 9(2): 3155-3194. [2]. Hill J L. Bayesian nonparametric modeling for causal inference[J]. Journal of Computational and Graphical Statistics, 2012.

Confidence in this Review

2-Confident (read it all; understood it all reasonably well)


Reviewer 5

Summary

This paper is about using observational data to get informative priors using a GPs with a non-stationary kernel. In cases where interventional data is limited, this method can be used to speed up learning as is shown in experimental results.

Qualitative Assessment

I think the two contributions of this paper are a) The idea of using the GP posterior over observational data as a prior and b) Using an affine transformation of this prior to model the interventional data. Both seem potentially useful in small data setting for interventional data as shown in non-trivial improvements for synthetic data sets.

Confidence in this Review

2-Confident (read it all; understood it all reasonably well)